# Metacommunity structure preserves genome diversity in the presence of gene-specific selective sweeps under moderate rates of horizontal gene transfer

Simone Pompei[1]*, Edoardo Bella[2], Joshua S. Weitz[3,4,5], Jacopo Grilli[6☯], Marco Cosentino Lagomarsino[1,2,7☯]*

**1** IFOM ETS - The AIRC Institute of Molecular Oncology, Milan, Italy, **2** Dipartimento di Fisica, Università degli Studi di Milano, via Celoria 16 Milano, Italy, **3** Department of Biology, University of Maryland, College Park, Maryland, United States of America, **4** Department of Physics, University of Maryland, College Park, Maryland, United States of America, **5** Institut de Biologie, École Normale Supérieure, Paris, France, **6** Quantitative Life Sciences, The Abdus Salam International Centre for Theoretical Physics (ICTP), Trieste, Italy, **7** I.N.F.N, via Celoria 16 Milano, Italy

☯ These authors contributed equally to this work.
* simone.pompei@ifom.eu (SP); marco.cosentino-lagomarsino@ifom.eu (MCL)

**Data Availability Statement:** The code used for the simulations is provided in the GitHub repository: https://github.com/eddbell/

## Abstract

The horizontal transfer of genes is fundamental for the eco-evolutionary dynamics of microbial communities, such as oceanic plankton, soil, and the human microbiome. In the case of an acquired beneficial gene, classic population genetics would predict a genome-wide selective sweep, whereby the genome spreads clonally within the community and together with the beneficial gene, removing genome diversity. Instead, several sources of metagenomic data show the existence of "gene-specific sweeps", whereby a beneficial gene spreads across a bacterial community, maintaining genome diversity. Several hypotheses have been proposed to explain this process, including the decreasing gene flow between ecologically distant populations, frequency-dependent selection from linked deleterious alleles, and very high rates of horizontal gene transfer. Here, we propose an additional possible scenario grounded in eco-evolutionary principles. Specifically, we show by a mathematical model and simulations that a metacommunity where species can occupy multiple patches, acting together with a realistic (moderate) HGT rate, helps maintain genome diversity. Assuming a scenario of patches dominated by single species, our model predicts that diversity only decreases moderately upon the arrival of a new beneficial gene, and that losses in diversity can be quickly restored. We explore the generic behaviour of diversity as a function of three key parameters, frequency of insertion of new beneficial genes, migration rates and horizontal transfer rates. Our results provides a testable explanation for how diversity can be maintained by gene-specific sweeps even in the absence of high horizontal gene transfer rates.

MetapopulationModel Simulated data and scripts used for the post-processing are provided in the Mendeley Data repository: https://data.mendeley.com/datasets/w7mdmtrtnp/1.

**Funding:** M.C.L. and SP were supported by Associazione Italiana per la Ricerca sul Cancro AIRC IG grant no. 23258. S.P. was supported by Fondazione Umberto Veronesi. JSW is supported in part by a grant from the Charles Blaise Pascal program of the Île-de-France region. The funders had no role in study design, data collection and analysis, decision to publish, or preparation of the manuscript.

**Competing interests:** The authors have declared that no competing interests exist.

## Author summary

Our study investigates the interactions between gene-specific selective sweeps, horizontal gene transfer (HGT), and genome diversity within microbial communities. Using mathematical models and simulations, we examine the time scales involved in these processes to gain insights into their complex dynamics. Our findings highlight the importance of considering both HGT and metacommunity structure. Contrary to previous assumptions that gene-specific selective sweeps lead to a decrease in biodiversity, we find that genome diversity can be regenerated under moderate rates of HGT. Our results offer a verifiable explanation for the preservation of diversity in the presence of gene-specific sweeps, even when high rates of horizontal gene transfer are absent.

## Introduction

Horizontal Gene Transfer (HGT) plays a crucial role in the processes that shape bacterial evolution [1–5]. HGT accelerates the adaptation of bacterial communities to new ecological niches [6, 7] and reduces the deleterious effects associated with the accumulation of genetic load by clonal reproduction [8]. HGT is also a widespread pathway through which pathogenic bacteria acquire resistance to antibiotics [7, 9].

According to classical population genetics theories [10], when the rate of HGT is low, then vertical inheritance is the main mechanism for the expansion of novel genetic variants. In such cases, the evolutionary dynamics is described by clonal evolution. In this case, because of linkage effects, transfer of a highly beneficial gene results in a drastic reduction of the diversity, as a consequence of the clonal expansion of the mutant carrying the gene, whose genome would "sweep" together with the beneficial gene. We can term this process a genome-wide selective sweep.

In contrast to this scenario, several lines of metagenomic evidence from communities of phylogenetically related strains [1, 11, 12], support an alternative scenario of gene-specific sweeps, an evolutionary dynamics where a beneficial gene can reach fixation across species or strains, without erasing diversity. Reconciling this scenario with standard population genetic models would require very high recombination rates [11] compared to standard direct measurements of such rates [13–15]. Hence, the consensus is that more complex mechanisms should be in place [1].

In the past years, several alternative hypotheses were put forward to reconcile the evidence of gene sweeps with the estimated values of the recombination rate [13–15]. A first mechanism was inspired by the evidence collected by Shapiro and coworkers [11], and was introduced by Polz and coworkers [16]. This hypothesis uses the observation that HGT rates between pairs of species decline rapidly, following an exponential pattern, as a function of their genetic distance [17, 18], an effect that leads to an effective HGT barrier between populations of different species/strains, provided that the intra-population rate of the genomic changes is faster than the inter-population HGT rate. Therefore diversity could persist in a metacommunity (a group of populations including different species/strains based on spatially separate patches) in presence of selective effects. Moreover, another crucial factor resides in the diversified selection experienced across separate patches, stemming from the inherent environmental heterogeneity. This dynamics could maintain diversity by favouring divergent species or strains in distinct geographical locales. A second mechanism, proposed by Niehus and coworkers [6], proposes that the combination of HGT rates close to realistic estimates [13–15] and a migration rate smaller than the typical selection rates of beneficial mutations, can lead to the fixation of beneficial

genes without the decrease of genome diversity. However, dynamic models based on this second mechanism predict gene-sweep times to be excessively short (ranging from months to years), and would require high HGT rates (based on current estimates of genetic transfer rates) to match the observation that timescales of horizontal gene-sweeps remain extremely short compared with phylogenetic timescales [18–20].

A different hypothesis for gene sweeping might be the so-called "soft sweeps" [21, 22], where widespread (e.g. neutral) recombination through HGT can generate a pool of standing variation (in our case promoting the presence of one or more beneficial alleles of a given gene across species) that is sufficient to support the emergence of multiple (interfering) sweeps in parallel upon a change of selective pressure. In this case the gene-specific sweep would consist of multiple parallel sweeps of a gene that previously spread neutrally by HGT onto diverse genetic backgrounds. However, this scenario cannot explain situations where a gene sweep originates from a gene that is not already initially present neutrally in many species. Additionally, this requires high HGT rates, as in the previous explanations.

Another mechanisms, put forward by Takeuchi and coworkers in 2015 [23], involves the linkage of the beneficial sweeping gene with widespread (species-specific) deleterious alleles, which would lead to negative frequency-dependent selection. This mechanism was shown to explain a gene-sweep dynamics in quantitative terms, provided that the basal recombination rate, (the spontaneous rate, not affected by selective pressure), is sufficiently low. The widespread linkage of the beneficial gene with deleterious alleles or more generally the presence of linked loci under negative frequency-dependent selection, does not have a simple explanation, but the authors speculate that it could be the consequence of ecological interactions between bacteria and viral predators [23], possibly supported by a "Kill the Winner" dynamics [24]. Interestingly, this mechanism can work with relatively low gene-transfer rates, and actually requires a low basal recombination rate. This is notable because it challenges the intuition (and the requirement of the previous models) that high recombination rates are necessary for gene-specific selective sweeps.

Here, we propose a complementary eco-evolutionary mechanism whose key ingredient is a metacommunity structure. Our approach is related to the classic population genetics perspective on selective sweeps in structured environments [25–27], where it is well known that under certain conditions, the effect of a selective sweep on the neutral variation of a subdivided population can be different from naive expectations. Specifically, we assume that the environment is characterized by the presence of multiple patches (*e.g.*, nutrient patches as in marine snow [28]) and that their physical separation is the main limitation to the spread, through genome migration or HGT, of beneficial genes. As we show, a metacommunity structure can preserve diversity during the fixation dynamics of a beneficial gene without requiring high recombination rates.

## Results

### Model components and terminology

We introduce an evolutionary model that describes the dynamics of the community-wide gene pool diversity in a spatially structured environment, supporting different species in the presence of gene sweeps. Our model describes a metacommunity where each species dominates in a single habitat (implying that intra-habitat dynamics are typically characterized by neutral fixation or selective sweep of a single species/strain). Accordingly, our model does not deal with intra-population diversity, and we will use the term diversity only to refer to pan-metagenomic diversity. In the following, we aim to describe a scenario where the intra-species genetic diversity is very limited (and restricted to neutral diversity only), so that all the

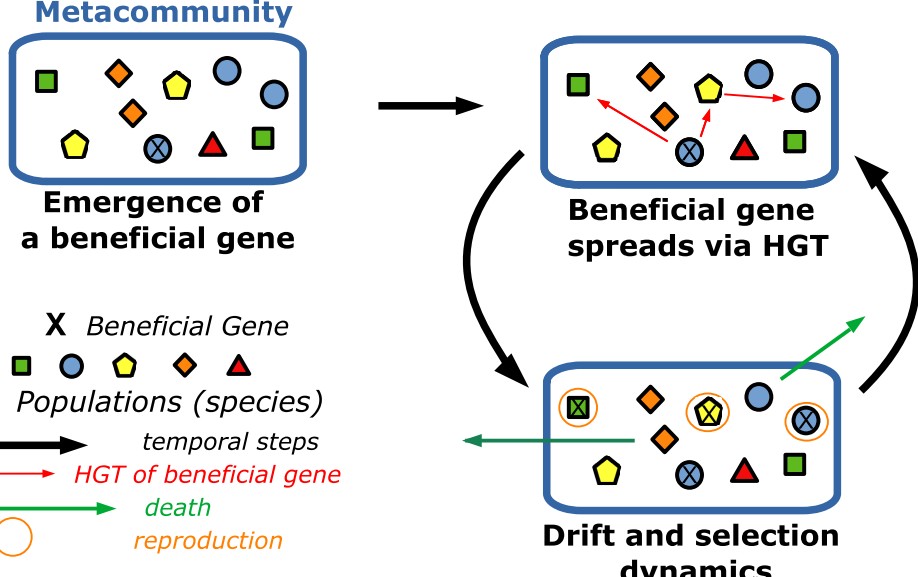

**Fig 1. Schematic representation of the multi-species metacommunity and the temporal dynamics of the model.** We consider a metacommunity consisting of *M* distinct patches (represented by symbols) and supporting different species (represented by colours/shapes). We assume that each patch is populated by only one species (a coloured geometric shape in the picture). Species in the metacommunity can contain a beneficial gene in their genome, here schematically represented by a cross. The time evolution of the metacommunity is the result of a migration-selection dynamics across patches, and selective effects are associated to the presence-absence of beneficial genes. The spread of beneficial genes in the metacommunity is the result of HGT events between patches.

individuals belonging to the same species can effectively be associated to a single "label" (the consensus genome of the individuals of the same species). We will further assume that all populations are connected, that migration of beneficial-gene carriers is the dominant process that reduces species diversity, and that the arrival of a beneficial gene in a patch, whether by horizontal transfer or by migration, is sufficient to guarantee a full sweep of the population.

Fig 1 illustrates the key model ingredients. We consider *M* distinct patches, each of which supports a single phenotypically homogeneous population, i.e., it only contains a single species. From a population genetics standpoint, this condition is similar to the evolutionary dynamics in the so called "periodic selection regime", i.e., an evolutionary scenario where the population is most of the time phenotypically homogeneous, with sporadic and fast selective sweeps of beneficial mutations [29–31]. This regime is (approximately) defined by the mathematical condition $\mu N \log N \leq 1$ (where $\mu$ is the beneficial mutation rate per generation, per individual and $N$ is the population size), which implies that at each generation there is (at most) one new emerging mutant. This condition is no longer valid for large population sizes (i.e., when $\mu N \log N \gtrsim 1$) or high mutation rates, since multiple beneficial mutations can emerge together and compete for fixation. In this evolutionary regime, called (in population genetics) "clonal interference" the population is no longer homogenous as multiple phenotypically distinct individuals co-exist within the same population. [32–36]. However, although the assumption of phenotypically homogeneous populations is no longer valid for in the clonal-infererence regime, our model still provides a lower bound estimate for the total biodiversity of the meta-community. Hence, the model predictions concerning the maintenance and regeneration of the diversity, i.e., whether or not the total diversity is removed during a gene-specific selective sweep, are valid also in the presence intra-population clonal interference effects.

In our model, the same species (corresponding, depending on the specific context, to ecotypes or strains) can populate more than a single patch. The metacommunity diversity is quantified by the number $S$ ($1 \leq S \leq M$) of distinct species present in the community at a given time-point. It should be noted that this definition is a standard measure of biodiversity used in the ecology related literature [37], but is not the standard diversity measure used in population genetics used to quantify the intra-population genetic variation.

A beneficial gene can spread across patches by two mechanisms, migration of an individual (and its genome), with the consequent genome-wide sweep of the strains carrying the gene, or HGT and sweep of the gene in the community. The first mechanism may reduce system-wide diversity, because the species in the invaded patch is replaced by the invader, while in the second scenario the beneficial gene is transferred across genetic backgrounds, with no loss of diversity In presence of a mechanism that maintains and regenerates diversity, new species are constantly generated with a neutral innovation rate $v$, hence giving rise to a typical time scale (of order $1/v$) over which the diversity loss from genome-wide sweeps may compete with the emergence dynamics of new species, inducing an increase of the diversity. In the model, as in standard Gillespie algorithmic approaches [38], time is counted in terms of the number of steps, and at the end of each step a single move will occur (e.g, a migration or an invasion). Hence, although each move is associated a physical rate, because of this normalisation, these physical rates can be treated as probabilities (i.e., are dimensionless). Time is also counted in terms of an another unit, which we will refer to as a "meta-generation" of the metacommunity, where 1 gen = $M$ time steps.

We note that we have chosen to adopt a generic terminology, such as "diversity-maintenance mechanism", "innovation", "invasion", "migration-sweep", "HGT-sweep", in order to make the model more flexible and conceptual. The terminology can be modified when thinking to specific experimental or real-world scenarios. For instance, when discussing a single species, innovation may refer to a large-scale mutational event or enough mutations to generate functional/phenotypic differences. In the case of multiple species or strains in a patchy environment, innovation is related to migration of unseen species/strains from distant patches or speciation.

## Dynamics of the diversity of the metacommunity in absence of HGT

In order to provide a mechanism for diversity-maintenance, we used a neutral process [37, 39–41] where the species occupation of patches change over time because of (i) neutral migration-substitution events, where a species occupying a patch is replaced by another existing species (ii) innovation events consisting in the emergence of a new species (which accounts for migration-invasion events from external species and speciation events).

Beneficial genes can be generated and spread across patches via HGT. We note that the specifics of the diversity-maintenance mechanism are not relevant for our results, which focus on the diversity loss (and time scale) due to the HGT-migration process of the beneficial gene. The only role of the diversity-maintenance process in our model is to provide a time scale that competes with the diversity loss due to gene sweeps. These characteristic time scales arise because different processes occur at different rates in the model. For example, the rate at which beneficial mutations occur and subsequently sweep through a population is different from the rate of neutral turnover of a population or the rate of invasion and subsequent take-over of a population. Purely on the basis of dimensional analysis, one can establish that such different rates give rise to different equilibration times, which in turn govern the different model regimes. In the following, we will explore three distinct regimes of the dynamics of the model, which correspond to different relative values of the competing time scales. Our main

result is that the diversity loss in presence of a beneficial gene can be moderate thanks to the presence of many patches supporting different species. Values of the rates associated to the different time scales will be defined and discussed in each regime.

We first focus on the evolutionary dynamics of the diversity-maintenance process, in the absence of HGT (Fig 2A), which will also be used as a "null" or "benchmark" case for our framework. This neutral model contains two elementary events: (i) neutral migration/sweep across patches and (ii) innovation events, corresponding in this context to the emergence of new species. At each time step, corresponding to the characteristic time scale of migration-sweep events, each population occupying a patch in the metacommunity can either change into a population corresponding to a species as a result of an innovation event (speciation or migration and neutral sweep from an outside pool of species), with a rate $v$ (per patch/per time

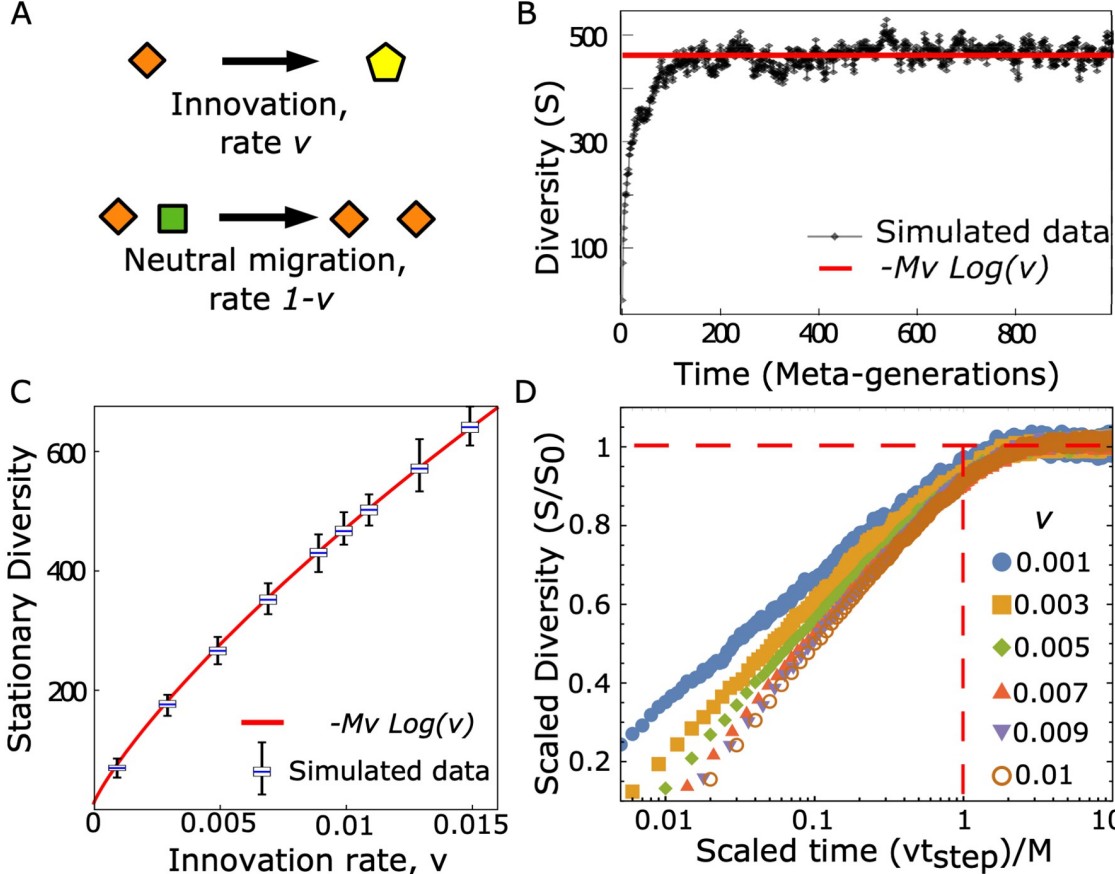

**Fig 2. In absence of HGT, an infinite-allele model provides a diversity-maintenance mechanism with an intrinsic time scale. A.** We consider a standard neutral model maintaining diversity. At each time step, corresponding to a basic migration-sweep time scale each patch can be swept neutrally by a new species, with an innovation event taking place with rate $v$ (per patch, per time step), or alternatively, its species can migrate and sweep, invading another patch (and sweeping neutrally), with a rate $1 - v$. Panel **B** shows diversity ($S(t)$, defined as the total number of distinct species present in the metacommunity at a given time $t$) in a typical simulation. Diversity relaxes to a plateau, in agreement with (Eq 1). The parameters of the simulation are $v = 0.01$ and $M = 10\,000$. **C.** Comparison between Eq 1 (valid in the limit $M \gg 1$ and $v \ll 1$) and simulated data for the equilibrium value of the diversity, plotted as a function of the innovation rate ($v$). Simulations correspond to $M = 10\,000$ and averages over 100 independent realizations. For each simulated value of $v$ the distribution of diversity $S$ is shown as a box plot (blue line: mean value, box: inter-quartile range, fences: max and min values) **D.** Characterization of the equilibration time scale for the average trajectories of the diversity. The plot shows diversity (scaled by its equilibrium value) vs time, scaled by the common equilibration time scale $N/v$. Simulations were performed over 100 independent realizations, for different values of the innovation rate $v$ (shown in the legend, coded by color and symbol). All the simulations were initialized with a metacommunity of a single species ($S(t = 0) = 1$).

step) or migrate and sweep neutrally to another patch, with a rate $1 - v$, (per patch/per time step) causing the extinction of the pre-existing species in the invaded patch (Fig 2A).

In this regime, the diversity $S$ displays equilibration dynamics (Fig 2B), reaching a stationary state $S_0$. The analytical expressions for the equilibrium value can be derived from classic calculations [37, 42, 43] (see also Methods), and is

$$\langle S_0 \rangle = -Mv \log(v) \; , \tag{1}$$

computed in the limit $M \gg 1$ and $v \ll 1$. Our numerical simulations of the model (Fig 2C) agree with Eq 1, showing that the typical equilibration time for this model setting is the inverse of the innovation rate $\tau_{eq} \simeq M/v$ steps $= 1/v$ gen (Fig 2D).

## Metacommunity diversity loss resulting from the introduction of a beneficial gene

We next ask how much diversity is lost upon the introduction of a beneficial gene, even in the absence of diversity-maintenance mechanisms. This section assumes only a single beneficial gene, and that no two beneficial mutations can simultaneously sweep at distinct loci. In order to address this question, we focused on the dynamics following the introduction of a beneficial gene that can spread through the metacommunity via both HGT and sweep (gene-specific sweep) and genome-wide sweeps on single patches. The initial diversity was set using the neutral model described in the previous section (hence depends on $v$, see Eq 1). We considered the limiting situation where no diversity-restoring mechanism was in action during the whole sweep. In other words, we assume that the fixation dynamics of the advantageous gene in the metacommunity is much faster than the equilibration time scale of the neutral biodiversity model. Roughly, if there are $N$ individuals per patch, this limit corresponds to the condition

$$v \ll \frac{1}{M \log N} \quad \text{and} \quad v \gg \frac{1}{MN}, \tag{2}$$

i.e., where the genome sweep time $M \log(N)$ is much smaller than $1/v$, but this is in turn much smaller than a metacommunity-wide fixation time of the neutral dynamics, which is order $MN$ (see Methods). This assumption, which will be relaxed in the next paragraph, is the most conservative scenario (the most adverse in terms of diversity loss) for the introduction of a new beneficial gene in a metacommunity.

Under these assumptions, only two processes take place (Fig 3A): (i) migration-sweep of a patch by a species carrying the beneficial gene (genome-wide sweep), which leads to a reduction of diversity, and (ii) spread of the beneficial gene by HGT (gene sweep), which does not reduce diversity. Hence, diversity can only decrease in this scenario, and we are interested in the magnitude of the decrease relative to the diversity baseline (see details in the Methods, Eq 7).

To implement genome-wide migration-sweep events, at each time step, with a rate $p_m$ (per patch, per time step), two patches are picked randomly. If the first patch carries the beneficial gene and the second one does not, the species of the second one is replaced by a copy of the first one. Similarly, HGT gene-sweep events occur at each time step with a rate $p_h$ (per patch, per time step). In such events, two random patches are selected. If the species of the first one carries the beneficial gene and the second one does not, then the gene is horizontally transferred and spreads into the second patch, without any displacement of species. HGT and migration rates are independent and we consider a fully connected network of communities with uniform rates (the spatial distance between patches is not modelled).

This model configuration corresponds to an evolutionary regime where the maintenance of diversity is slow compared to the time scale of fixation dynamics. For this regime we find (see

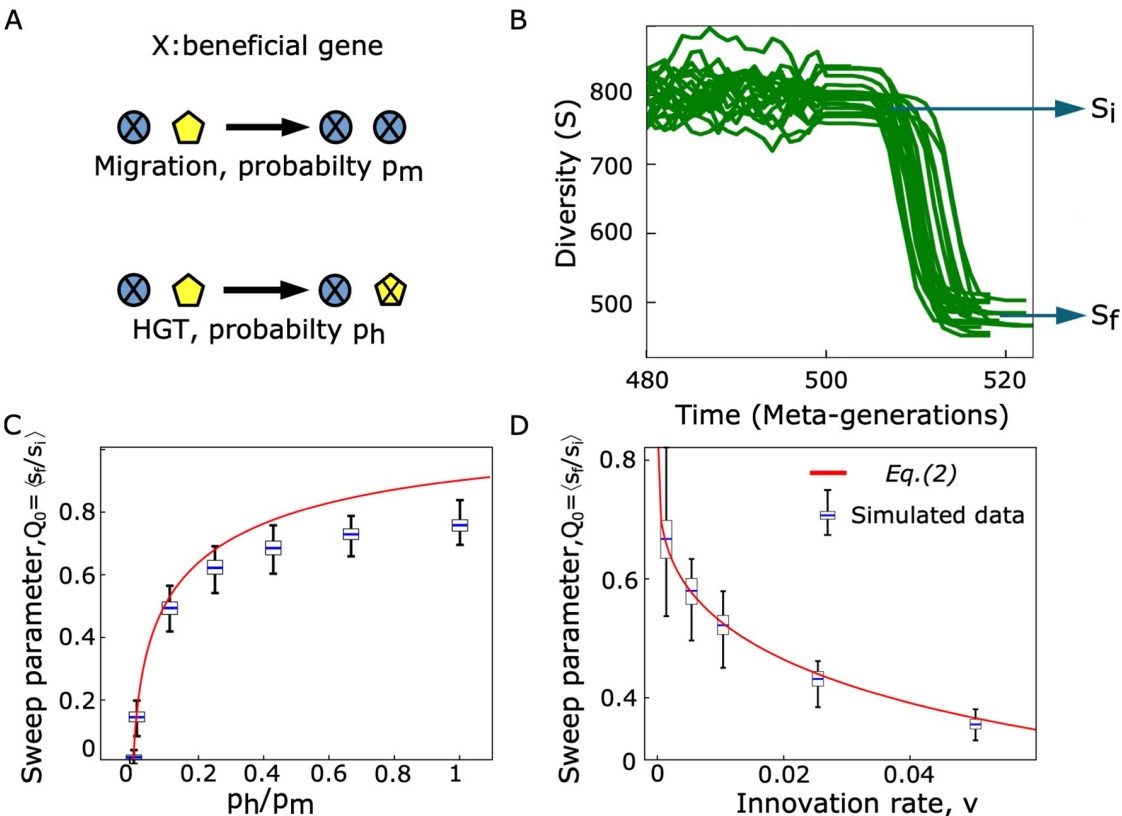

**Fig 3. In presence of HGT only, and no diversity-maintenance mechanism, the fixation of a beneficial gene in the metacommunity can lead to a moderate loss of diversity. A**. A beneficial gene can spread across patches (i) via migration events (reducing diversity) or (ii) via HGT-sweep (maintaining diversity) The two processes take place at each time step, with rate $p_m$ (per patch, per time step) and $p_h$ (per patch, per time step) respectively. Here, we assume that no diversity-maintenance mechanism counteracts the diversity loss (this assumption will be relaxed later) **B**. Simulations of this model show a diversity ($S(t)$) loss from the initial value ($S_i$) to a new stable value ($S_f$), corresponding to complete invasion of the beneficial gene. Each solid line is a realization, with initial condition of the simulations generated by the neutral model described in [Fig 2] ($M = 10000$ and $v = 0.02$), and with parameters $p_h = 0.2$ and $p_m = 1 - p_h$. **C-D**. Comparison between analytical prediction ([Eq 3]) and simulated data for the sweep parameter $Q = \langle S_f / S_i \rangle$ as a function of the ratio $p_h/p_m$ (panel **C**, $v = 0.01$), and of the innovation rate of the neutral model generating the initial diversity (panel **D**, $p_h = 0.1$, $p_m = 0.9$). Simulations performed with $M = 10\,000$. The panels **CD** show the distribution of diversity $S$ over 100 realizations as a box plot (blue line: mean value, box: inter-quartile range, fences: max and min values).

Methods) that the number of populations (patches) carrying the beneficial gene follows a logistic growth (see Methods, Eq 11) and after a time $\tau_{\text{fix}} \simeq \frac{2M \log(M)}{p_h + p_m}$ steps $= \frac{2 \log(M)}{p_h + p_m}$ gen the diversity reaches an absorbing state where all the species in the metacommunity carry the advantageous gene. Mathematically, this evolutionary regime is defined by the condition $\tau_{\text{fix}} \ll \tau_{\text{eq}}$ (see Methods). The key aspect is the residual value of the diversity after the fixation of the beneficial gene.

Fig 3B shows an example of the typical dynamics of the diversity after the introduction of the beneficial gene. The initial value of the diversity ($S_i \simeq \langle S_0 \rangle$) decreases after the introduction of the beneficial gene and reaches a new stationary value $S_f$. We quantify the effect of the fixation of the beneficial gene by the "sweep parameter" $Q_0 \equiv \langle \frac{S_f}{S_0} \rangle$. Thus, by definition, a value of $Q_0 \simeq 0$ corresponds to a scenario of a genome-wide sweep across the metacommunity, while for $Q_0 > 0$ some diversity is regenerated.

We have derived an approximate analytical solution for the model dynamics in this regime (i.e., when $\tau_{\text{fix}} \ll \tau_{\text{eq}}$, see Methods), which leads to the following expression for the sweep

parameter,

$$Q_0 = 1 - \frac{\log\left(1 - (1-v)e^{-\frac{p_h}{p_m}}\right)}{\log(v)} \ . \tag{3}$$

Eq (3) was derived under the assumptions of small $v$, small $p_h$, large metapopulation size $M \gg 1$ (see Methods), and is in good agreement with numerical simulations of the model (Fig 3C and 3D). These results show that a full genome-sweep dynamics ($Q_0 = 0$) can only be reached when $p_h/p_m \to 0$, i.e., when the HGT rate is completely negligible (e.g., for $p_h \to 0$ or $p_m \gg p_h$) and the "invasion" dynamics of the species with the beneficial gene is the only relevant one. However, as soon as the HGT rate is non-negligible (for any positive value $p_h/p_m > 0$), there is more than a single species within the metacommunity after the fixation of the beneficial gene. More specifically, for values of $p_h/p_m \simeq 0.1$ and $v = 0.01$, we already obtain $Q_0 \simeq 0.5$, which means that a HGT-sweep rate ten times slower than the typical migration-sweep time is sufficient to regenerate (in the worst-case scenario) half of the diversity within a metacommunity. We note that the selection coefficient for beneficial mutations does not play a role here in the expressions of Eqs 3 and 4, as we have assumed that any carrier of the beneficial gene will sweep a patch.

## Gene-sweep dynamics under competing time scales

Having quantified how a metacommunity may preserve diversity under a gene sweep in the absence of diversity-restoring mechanisms (i.e., when $\tau_{\text{fix}} \ll \tau_{\text{eq}}$), we now study how diversity and multiple rounds of gene sweep may interact over longer time scales. Specifically, we consider a regime where the emergence and fixation dynamics of beneficial genes takes place on a time scale that is comparable with the equilibration time of the diversity-restoring mechanism, which occurs when $\tau_{\text{fix}} \gtrsim \tau_{\text{eq}}$ and is realized in our case as a neutral model (Fig 4A). In this regime, three different evolutionary forces are acting at each time step: (i) innovation events, taking place at a rate $v$ (per patch, per time step), (ii) migration of a species with the beneficial gene into a patch that did not carry the gene (rate $(1 - v)p_m$, per patch, per time step), and (iii) transfer of a beneficial gene via HGT and sweep (rate $(1 - v)p_h$ per patch, per time step). In order to fully specify the model, we need to state how likely new species carry a beneficial gene. We assume that in an innovation event, the new species carries the beneficial gene with a probability $f_0(t) = D_s(t)/M$, i.e., equal to the fraction of populations (patches) carrying the beneficial gene at the time of the event. This assumption is justified when innovation represent migration events from a parallel metacommunity where the beneficial gene has the same frequency across patches. Furthermore, this assumption would hold true if innovation arose from neutral mutations, occurring with equal probability on any genetic background (i.e., equally among species possessing the beneficial gene and those without it). Finally, a species can migrate and sweep to another patch with its same genetic content (both species with the beneficial gene, or both without the gene) with a neutral rate $1 - v$ (per patch, per time step).

Fig 4B shows the typical dynamics of diversity after the introduction of a single beneficial gene. Diversity drops to a minimum value ($S_{min}$), in a time-scale $\tau_{\text{fix}} \simeq \frac{2M \log(M)}{p_h + p_m}$ steps $= \frac{2 \log(M)}{p_h + p_m}$ gen, for $v \ll 1$, similar to the case of the time-separation limit (see Methods). However, in this case, the innovation dynamics restores the initial value of the diversity, on a time scale $\tau_{\text{eq}} \simeq M/v$ steps $= 1/v$ gen. Moreover, because of the generation of new species during the fixation dynamics, the minimum value of the diversity is typically

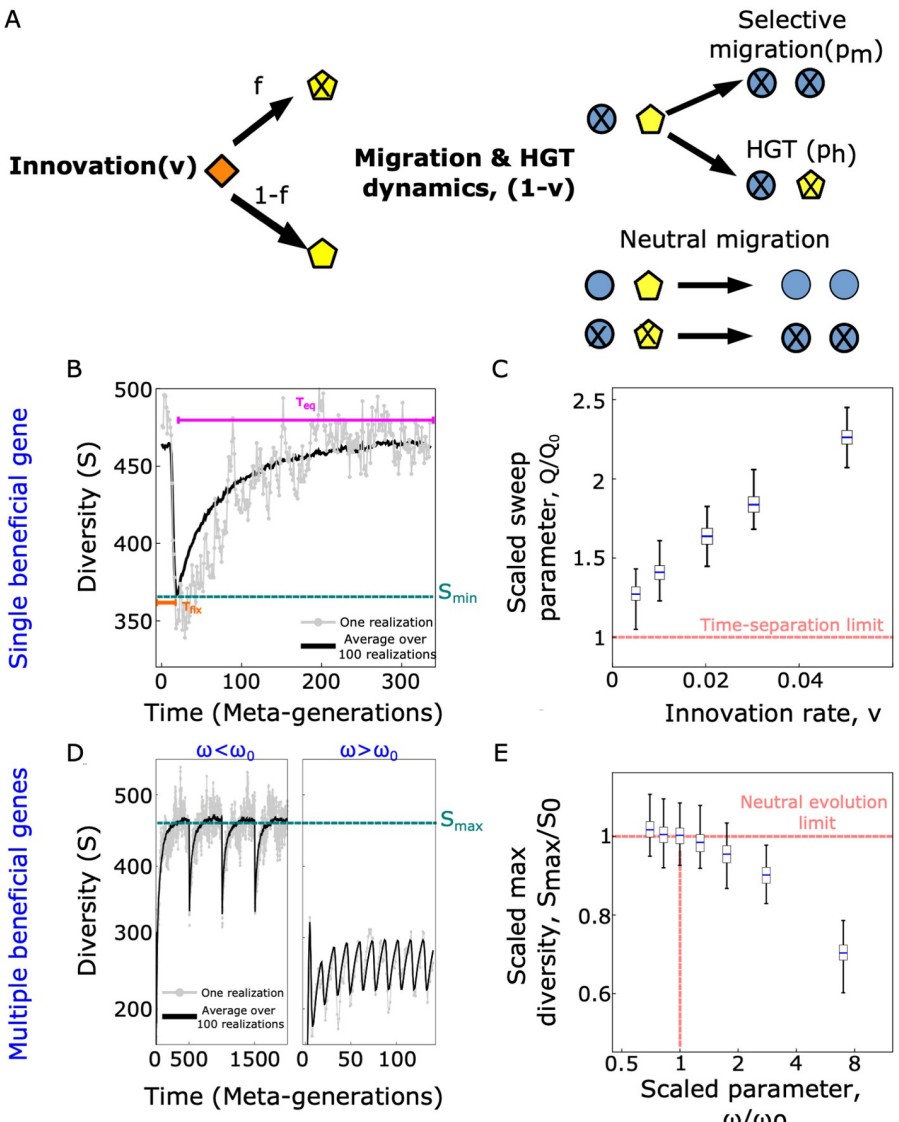

**Fig 4. Competition of gene-sweep and diversity-restoring dynamics affects the minimal and maximal observed diversity. A**. Spread of a beneficial gene over a metacommunity can compete with a diversity-restoring mechanism occurring with a rate $v$. HGT-sweep and migration-sweep events take place with a joint rate $1 − v$ and are realized by picking two random patches within the metacommunity. In innovations, new species will carry a beneficial gene with probability $f_0$ equal to the fraction of populations (patches) within the metacommunity carrying the beneficial gene. Neutral migration-sweep events occur if both populations carry the beneficial gene, or neither of them does. If the first of the two populations carries the beneficial gene and the second does not, a (selective) migration-sweep event occurs with probability $p_m$ (and a total rate $(1 − v)p_m$) while an HGT-sweep event occurs with probability $p_h$ (and a total rate $(1 − v)p_h$), see [Fig 3](). **B**. The dynamics of diversity after the emergence of a beneficial gene is characterized by two time scales: (i) the time until the beneficial gene reaches fixation ($\tau_{fix}$), during which the diversity drops and (ii) an equilibration time ($\tau_{eq}$), restoring its initial value. **C**. Minimum diversity quantified by the scaled sweep parameter $Q/Q_0$, where $Q = \langle S_{min}/S_0 \rangle$, and $Q_0$ is the sweep parameter ([Fig 3]()). The distribution of $Q/Q_0$ shown as a box plot shows an increasing trend with increasing innovation rate $v$. **D**. If multiple beneficial genes emerge periodically after a time $1/\omega$, diversity shows an oscillatory dynamics. **E**. The maximum diversity ($S_{max}$, shown as a box plot) divided by the expected value under neutral biodiversity ($S_0$), decreases after a critical value of the scaled parameter $\omega/\omega_0$, where $\omega_0^{-1} = \tau_{fix} + \tau_{eq}$). Other model parameters: $M = 10000$, $p_h = 0.1$, $p_m = 0.9$. All rates are per patch, per time step unless otherwise specified.

higher than the one observed in the time-separation limit, that corresponds to the mathematical limit $\nu \to 0$ and occurs when $\tau_{\text{fix}} \ll \tau_{\text{eq}}$ (Fig 3). We have quantified this effect with numerical simulations of the model, using the sweep rate, now defined as

$$Q = \left\langle \frac{S_{min}}{S_0} \right\rangle. \tag{4}$$

Fig 4C shows that for innovation $\nu = 0$ one obtains the same minimum for the diversity as in the limit case without any diversity-restoring mechanism, $Q = Q_0$. Conversely, the minimal diversity is always higher than in absence of a diversity-restoring mechanisms ($Q > Q_0$) under a positive innovation rate $\nu > 0$. These observations arise due to the competition between the time scale of the diversity drop via the sweeping gene and the time scale of the diversity-restoring mechanism. If the restoring mechanism is sufficiently fast, the diversity-drop mechanism does not have time to reach its natural minimum value (due to the complete sweep) observed in Fig 3B.

Due to the same competition of time scales, multiple acquisitions of beneficial genes may have consequences on the *maximal* diversity observed in our model. The mechanism is illustrated by Fig 4D and 4E. We have considered a regime where multiple beneficial genes arrive in the metacommunity with a constant frequency $\omega$ (per time step). Additionally, we assumed a fitness scenario where the last-emerged gene always carries the highest beneficial effect. In a simple simulation where beneficial genes arrive periodically after a time $1/\omega$ the resulting dynamics of the diversity shows an oscillatory pattern due to successive gene-sweeps (Fig 4D), with oscillations corresponding to rounds of gene-sweep and diversity-restoring dynamics. For beneficial genes arriving stochastically at a constant rate $\omega$, the oscillations disappear, but the average diversity display similar behaviour (S1 Fig).

If the rate of arrival of new beneficial genes is too high, the diversity-restoring mechanism does not have enough time to achieve its steady-state diversity. This process is illustrated by Fig 4E, which quantifies the maximal diversity as a function of the arrival rate of new beneficial genes. We can define a frequency $\omega_0^{-1} := \tau_{\text{fix}} + \tau_{\text{eq}}$. In case $\omega^{-1} < \omega_0^{-1}$, the time scale for the emergence of the beneficial rate is faster than the equilibration dynamics, hence the diversity cannot be restored to its initial value. The difference between these two regimes is visualized by the two sub-panels of Fig 4D and leads to the consequences for the diversity quantified by Fig 4E.

S1 Table recapitulates the four different regimes for the qualitative behaviour of the diversity in terms of two dimensionless quantities, $\omega(\tau_{\text{fix}} + \tau_{\text{eq}})$, and $\tau_{\text{fix}}/\tau_{\text{eq}}$. The first quantity compares the typical time between arrivals of the beneficial genes with the total time needed to fix them and to equilibrate the system by its neutral dynamics. The second is the ratio of the fixation time of the beneficial gene with the neutral equilibration time (which sets the dominant dynamics).

## Conclusion

We have shown that the simplified framework of our eco-evolutionary model with an underlying metacommunity structure, can support a "gene sweep" dynamics, without eliminating genome diversity. Instead, gene sweeps can lead to a moderate reduction of diversity even in the absence of diversity-restoring mechanism. Inclusion of a diversity-restoring mechanism (e.g, a neutral biodiversity model in our specific case) can increase the minimal observed diversity. Conversely, for high rates of beneficial mutations, it could lead to a reduction in the maximum diversity. The mechanism by which a metacommunity maintains diversity under a gene sweep is compatible with small HGT rates compared to typical migration time scales. Unlike

prior work, our model does not explicitly require additional ingredients such as frequency-dependent selection at the individual level, induced by genome-level processes or by ecological interactions.

Most of the limitations of our model come with a trade-off with its simplicity. Specifically, numerous oversimplifications, including the absence of spatial organization, no relationship between migration and HGT rates, and a highly simplistic approach to intra-population dynamics, present intriguing questions for future research endeavours. For example, in our model we considered exclusively neutral non-beneficial mutations, and we did not include deleterious mutations. The presence of such deleterious mutations could potentially diminish overall genetic diversity and introduce the possibility of modifying sweep timescales through linkage effects. Similarly, we did not investigate clonal interference effects. In addition, the model outcomes rely on simple time-scale competition arguments. From this standpoint, our hypothesis is related in spirit to the classic proposition put forward by G. A. Hutchinson [44] to justify the very high observed microbiological diversity in samples of ocean pythoplankton (which was at odds with the principle of competitive exclusion, according to which the survival of a single species within a population should be privileged). To reconcile diversity with competitive exclusion, Hutchinson argued that if the time scale at which the exclusion principle is enforced were comparable to the time scale over which environmental conditions change significantly, a state of equilibrium would never be reached, and therefore there would be no predominance of a single species.

Our focus on a metacommunity is complementary to the approach assumed by the previous study by Niehus and coworkers [6], which focused on intra-patch diversity of a *single* population and the role of migration of non-carrier individuals, favouring diversity in moderate amounts. The same study also showed that such migration effects are enhanced in a small metacommunity, made of multiple patches. In our model, all populations are connected, and the within-population dynamics are assumed to be fast and result in a single winner. More precisely, we took the conservative assumptions that (i) migration of carriers (which reduces the diversity) is the dominant process and that (ii) the presence of the beneficial gene on a patch, whether it is carried by a species invading the patch by migration or if is acquired by HGT, is sufficient to guarantee a full sweep of the population. Phenomena akin to those described by Niehus and colleagues would increase the prediction of the residual diversity in our model. Thus, in light of their study, we can consider our estimates as lower bounds for diversity.

Apart from these differences, the mechanisms described by our work are conceptually similar to the ones discussed by Niehus and coworkers, in that they are a result of the balance between HGT and migration rates. As noted in ref. [6], these mechanisms lead to an *effective* frequency-dependent selection (which in our case acts completely at the level of populations within a metacommunity, not on individuals), as it reproduces the same effect defined by Takeuchi and coworkers [23]. However, we note that this dependency has a different origin than the processes hypothesized by Takeuchi and coworkers [23] (which act at the level of an individual within a population). In the scenario assumed by Takeuchi and colleagues, the diversity is favoured by ubiquitous and diverse deleterious loci that are linked to the acquired beneficial gene. In such cases, the diversity of bacterial species should be capped by the number of deleterious linked effects (e.g. phage diversity). If these linked alleles can be quantified in data, they should be linked to residual diversity after a gene sweep.

Importantly, even though they are conceptually different, these hypotheses are not mutually exclusive, and possibly can both be detected in data or addressed in controlled experiments. To address negative frequency-dependent selection due to linkage, genomic analysis of

microbial communities undergoing gene sweeps should be able to isolate the linked deleterious loci that co-occur with the beneficial gene in each species or strain. In order to test the role of a patchy community in restoring diversity, experiments could induce gene sweeps in a laboratory metacommunity with varying densities of patches. In such a setting, spatial patterns of the frequency spectrum or the genetic diversity, intended as the two-point measures of diversity related to the variance of allele frequency could be compared to the variance of the number of co-existing species in the metacommunity predicted by different models (see S2 Fig). More specifically, the key observables would be statistics of the observed polymorphisms, as the central point is to identify the presence of selective forces acting on genomic regions other than the one embedding the favoured gene. Additionally, the model predicts how migration effects may affect residual diversity in a gene sweep, and this could be possible controlled in such "laboratory gene sweep" setups. Experimental systems that might allow this are conceivable today [28], although complex spatio-temporal processes might complicate considerably the experimental scenario compared to the simple, purely conceptual, model proposed here [45, 46]. Despite these limitations, future studies of genomic data may be able to differentiate a gene-specific sweep with or without high HGT based on an analysis of additional selective forces in other portions of the genome.

## Materials and methods

### Within-patch and between-patches fixation dynamics

Let us first consider an individual patch supporting the growth of $N$ clonal individual. Let us fix the timescales so that the generation rate (time) equals to 1. We consider a mutation with selective advantage $s$ and assume $1/N \ll s \ll 1$. The fixation probability of such a mutation equals $1 - \exp(-s) \approx s$, and the typical fixation timescales (intra-patch sweep time) equals $\tau_{\text{patch}} \sim 1/s \log(Ns)$ (see e.g. [30, 31]).

Successful migrations and HGT of a beneficial genome (gene in the case of HGT) with selective advantage $s$ occurs with rates $p_m$ and $p_h$ respectively. Assuming that the timescales of intra- and inter- patch dynamics are separated, these rates can be decomposed into two contributions: a basal migration / HGT rates, equal to $\mu_m$ and $\mu_h$ respectively, and the fixation probability equal to $s$. Therefore we have $p_m = s\mu_m$ and $p_h = s\mu_h$.

The other processes shaping the communities are the innovations and the displacement of a species by another one from a patch due to neutral migration. We define $\mu_I$ as the rate of arrival of a new species in the patch. Assuming neutrality, the probability of fixation in a patch is $1/N$. We have therefore that a new species is introduced with rate $\mu_I/N$ and the displacement of a species on a patch via migration occurs with rate $\mu_m/N$.

In all our derivations we assume a time-scale separation between these three processes: fixation of beneficial mutations in a single patch (with rate $\approx s/\log(Ns)$), migration/HGT across patches (with rates $s\mu_m$ and $s\mu_h$), and neutral processes and diversity innovations (rates $\mu_m/N$ and $\mu_I/N$):

$$s/\log(Ns) \gg s\mu_m \approx s\mu_h \gg \mu_m/N \approx \mu_I/N \ . \tag{6}$$

Together with the assumption $1/N \ll s \ll 1$, this condition impose the constraints

$$\mu_I \approx \mu_m \approx \mu_h \ll 1/\log(Ns) \ , \tag{6}$$

where time is measured in generations (here defined as the typical doubling time of individuals within a patch). We note that the ratio $p_m/p_h$ in our model should not become too small or too large, in order for this time-separation assumption to be fully consistent.

## Expected diversity for the neutral model

This section focuses on a model configuration where only two evolutionary forces are present: (i) innovation events, taking place at rate $v$ (per patch, per time step) and (ii) neutral migrations (rate $1 - v$, per patch, per time step). This configuration corresponds to the standard Hubbell's model [37]. In the following, we will show that, using analytical results known for such model classes [37, 42], one can easily compute the expected value of the diversity at equilibrium, in the neutral scenario (Eq 1).

Hubbel's model can be mapped into a urn process, where each patch is represented by a ball with a color corresponding to its species, and it can be shown [37, 42] that the distribution of the species abundance at equilibrium is given by Ewens's sampling formula, for the distribution of alleles under neutral mutations [43], in the context of population genetics

$$\langle S_M(n) \rangle = \frac{\theta}{n} \frac{\Gamma(M+1)\Gamma(M+\theta-n)}{\Gamma(M+1-\theta)\Gamma(M+\theta)}, \tag{7}$$

where $\Gamma(x) = \int_0^\infty t^{x-1} e^{-t}\, dt$ is the standard Gamma function.

This distribution defines the expected number of species occupying exactly $n$ patches, and is specified in terms of the model parameter $\theta = \frac{(M-1)v}{1-v}$, called the fundamental biodiversity number. The expected number of species co-existing in the metacommunity at equilibrium is then obtained by summing over all the elements of this distribution [42]

$$\langle S_0 \rangle \equiv \sum_{k=1}^{M} \langle S_N(k) \rangle = \sum_{i=0}^{M-1} \frac{\theta}{\theta + i}, \tag{8}$$

In the limit $M \gg 1$, we can replace the sum with an integral,

$$\langle S_0 \rangle \simeq \int_0^{M-1} \frac{\theta}{\theta + x} dx = \theta \log\left(\frac{M-1+\theta}{\theta}\right) = \frac{(M-1)v}{1-v} \log\left(\frac{1}{v}\right). \tag{9}$$

In the limit of small values of $v$ and by replacing $M - 1 \rightarrow M$, we obtain the approximated solution of Eq (1)

$$\langle S_0 \rangle = -Mv \log(v). \tag{10}$$

## Model dynamics during the spread of a beneficial gene, without diversity-maintenance mechanism

In this section, we focus on the model variant described in Fig 2, where a beneficial gene is introduced in the metacommunity and can spread through to invasions and HGT events, with a total rate $p_h + p_m$ (per patch, per time step), and derive an approximated analytic solution of the model dynamics.

First, we focus on the the time-evolution of the average number of patches with populations carrying the beneficial genes $(B(t))$, for which we consider the deterministic limit (i.e., without considering stochastic fluctuations) described by a logistic growth

$$\begin{cases} \dfrac{d}{dt} B(t) = (p_m + p_h)\left(\dfrac{B(t)}{M}\right)\left(\dfrac{M - B(t)}{M}\right) \\ B(0) = 1 \end{cases}, \tag{11}$$

where time is measured in time steps from the introduction of the beneficial gene, in the

continuous limit approximation. The expected time to the fixation of the beneficial gene in the metacommunity ($\tau_{\text{fix}}$) can be computed using the solution of Eq 11, which reads

$$B(t) = \frac{Me^{\frac{t(p_h + p_m)}{M}}}{e^{\frac{t(p_h + p_m)}{M}} + M - 1} \tag{12}$$

and imposing the condition $B(\tau_{\text{fix}}) = M - 1$, which is fulfilled by

$$\tau_{\text{fix}} = \frac{2M \log(M - 1)}{p_m + p_h} \simeq \frac{2M \log(M)}{p_m + p_h}, \tag{13}$$

where time is counted in time steps.

Next, we focus on the extinction dynamics of a species without the beneficial gene. To compute the extinction probability of such species, we start by considering the dynamics of the number of patches without the beneficial gene and with species $s$ ($D_s(t)$). In the model configurations considered in this context, this class of species can only decrease over time because of the invasion of species carrying the beneficial gene. The time evolution of $D_s(t)$ and given by

$$\begin{cases} \dfrac{d}{dt}D_s(t) = & -p_m \left(\dfrac{B(t)}{M}\right)\left(\dfrac{D_s(t)}{M}\right) \\[2ex] D_s(0) = & M_s^0, \end{cases} \tag{14}$$

since invasion events occur with a rate $p_m$, and cause a reduction of $D_s(t)$ if one of the two populations picked randomly belongs to $s$ (i.e., chosen with probability $D_s(t)/M$) and the other one carries the beneficial gene (i.e.,chosen with probability $B(t)/M$). Here, time is measured in time steps from the introduction of the beneficial gene, in a continuous limit approximation. The solution to Eq (14) reads

$$D_s(t) = M_s^0 \left(\frac{M}{e^{\frac{t(p_h + p_m)}{M}} + M - 1}\right)^{\frac{p_m}{p_h + p_m}}. \tag{15}$$

Similarly, the probability that, at time $t$, one of the species $s$ acquires the beneficial gene by HGT is given by the product of the HGT rate, $p_h$ and the probability that one of the two populations picked randomly belongs to $s$ (i.e., chosen with probability $D_s(t)/M$) and the other one carries the beneficial gene (i.e.,chosen with probability $B(t)/M$)

$$p_s^{gain}(t) = p_h \left(\frac{B(t)}{M}\right)\left(\frac{D_s(t)}{M}\right) \tag{16}$$

$$= M_s^0 p_h e^{\frac{t(p_h + p_m)}{M}} \left(-e^{\frac{t(p_h + p_m)}{M}} - M + 1\right)^{-\frac{p_m}{p_h + p_m} - 1} (-M)^{-\frac{p_h}{p_h + p_m}}. \tag{17}$$

The expected number of species $s$ that, by time $t$, have gained the beneficial gene is

$$G_s(t) = \int_0^t p_s^{gain}(\tau)d\tau \tag{18}$$

$$= M_s^0 \frac{p_h}{p_m}\left(1 + M(-M)^{-\frac{p_h}{p_h+p_m}}\left(-e^{\frac{t(p_h+p_m)}{M}} - M + 1\right)^{-\frac{p_m}{p_h+p_m}}\right), \tag{19}$$

and the expected number of species $s$ that have gained the beneficial gene at any point in time can be computed as

$$G_s^{tot} = \lim_{t\to\infty} G_s(t) = M_s^0 \frac{p_h}{p_m}. \tag{20}$$

The probability of extinction for species $s$ can be computed assuming a Poisson distribution for the number of species $s$ that have gained the beneficial gene. This distribution has mean value $G_s^{tot}$, Hence, the probability to have gained 0 genes (i.e., to became extinct) reads

$$P_s^{ext}(M_s^0) = e^{-G_s^{tot}} = e^{-M_s^0 \frac{p_h}{p_m}}. \tag{21}$$

Finally, the sweep parameter, i.e., expected reduction of the biodiversity after the fixation of the beneficial gene, is obtained by averaging $P_s^{ext}(M_s^0)$ over the probability distribution of the number of patches populated by the same species ($P(M^0)$)

$$\left\langle \frac{S_f}{S_0} \right\rangle = 1 - \sum_{M^0=1}^{M} P(M^0) P_s^{ext}(M_s^0) \tag{22}$$

The probability distribution $P(M^0)$ is the normalized version of [Eq (7)](), and, in the limit of large metacommunity $M \gg 1$ and small innovation rate $\nu \ll 1$, is well approximated by the Fisher log series [42, 47]

$$P(M^0) \simeq -\frac{1}{\log(\nu)}\frac{(1-\nu)^{M^0}}{M^0}. \tag{23}$$

Under these assumptions, the sweep parameter can be computed analytically and reads

$$Q \equiv \left\langle \frac{S_f}{S_0} \right\rangle = 1 - \sum_{M^0=1}^{M} P(M^0) e^{-M^0 \frac{p_h}{p_m}}$$

$$\overset{M \gg 1}{\simeq} 1 - \frac{\log\left(1 - (1-\nu)e^{-\frac{p_h}{p_m}}\right)}{\log(\nu)}. \tag{24}$$

## Supporting information

**S1 Fig. Dynamics of the average diversity when beneficial genes emerge at a constant rate.**
**A**. We show here two examples of simulations use to investigate the dynamics of the diversity in the same model regime as [Fig 4](), when beneficial genes emerge stochastically at a constant

rate $\omega$ (values of $\omega$ specified above each plot, in terms of $\omega_0^{-1} = \tau_{\text{fix}} + \tau_{\text{eq}}$)). In this case no oscillations are observed, and the dynamics of the diversity is captured by its average value $S_{mean}$.**B**. The average diversity ($S_{mean}$, shown as a box plot) divided by the expected value under neutral biodiversity ($S_0$) shows a monotonic decrease with increasing value of $\omega/\omega_0$, where $\omega_0^{-1} = \tau_{\text{fix}} + \tau_{\text{eq}}$). Other model parameters: $M = 10000$, $p_h = 0.1$, $p_m = 0.9$. All rates are per patch, per time step unless otherwise specified.
(PDF)

**S2 Fig. The variability of the number of co-existing species in the metacommunity quantifies bio-diversity consistently with the mean value.** We show here the mean value and the standard deviations of the (i) diversity, defined as the number of distinct species co-existing in the metapopulation (panels **A,B** and **C**) and (ii) minimum ($S_{min}$)) and maximum ($S_{max}$) value of the diversity (panels **D,E**), evaluated in the dynamical regimes investigated in our model. Simulated data (and corresponding model parameters) used for this analysis are the same one displayed in the main figures, and are specified above each plot.
(PDF)

**S3 Fig. Breakdown of the linear analytic regime.** Panels previously displayed in Fig 2C (**A**), Fig 3C (**B**) and Fig 3D (**C**) are presented here on a logarithmic x-scale.
(PDF)

**S1 Table. Expected values of diversity in all model regimes.** This table summarizes the different regimes of the expected value of the biodiversity as a function of two dimensionless variables $\omega(\tau_{\text{fix}} + \tau_{\text{eq}})$, and $\tau_{\text{fix}}/\tau_{\text{eq}}$, including the three main time scales of the model: (i) the equilibration time of the system ($\tau_{\text{eq}}$), (ii) the fixation time of the beneficial gene ($\tau_{\text{fix}}$) and (iii) the time between arrival of beneficial genes ($\frac{1}{\omega}$). In each regime we illustrate the (i) expected maximum and minimum biodiversity ($S_{max}$ and $S_{min}$, expressed in terms of the expected neutral value of the diversity $S_0$ (Eq 1) and of the sweep parameter $Q_0$ (Eq 3) and (ii) the Figure showing the corresponding numerical results.
(PDF)

## Acknowledgments

We would like to thank Kunihiko Kaneko for useful discussions.

## Author Contributions

**Conceptualization:** Joshua S. Weitz, Jacopo Grilli, Marco Cosentino Lagomarsino.

**Formal analysis:** Simone Pompei, Jacopo Grilli, Marco Cosentino Lagomarsino.

**Investigation:** Simone Pompei, Edoardo Bella, Joshua S. Weitz, Jacopo Grilli, Marco Cosentino Lagomarsino.

**Methodology:** Simone Pompei, Edoardo Bella, Joshua S. Weitz, Jacopo Grilli, Marco Cosentino Lagomarsino.

**Project administration:** Marco Cosentino Lagomarsino.

**Software:** Edoardo Bella, Jacopo Grilli, Marco Cosentino Lagomarsino.

**Supervision:** Simone Pompei, Jacopo Grilli, Marco Cosentino Lagomarsino.

**Writing – original draft:** Simone Pompei, Joshua S. Weitz, Jacopo Grilli, Marco Cosentino Lagomarsino.

**Writing – review & editing:** Simone Pompei, Joshua S. Weitz, Jacopo Grilli, Marco Cosentino Lagomarsino.

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
