## [Decision Letter · Decision Letter 0]

7 Mar 2023

Dear Dr. Pompei,

Thank you very much for submitting your manuscript "A Metacommunity Scenario Preserves the Diversity in Presence of Gene-specific Selective Sweeps." for consideration at PLOS Computational Biology.

As with all papers reviewed by the journal, your manuscript was reviewed by members of the editorial board and by several independent reviewers. In light of the reviews (below this email), we would like to invite the resubmission of a significantly-revised version that takes into account the reviewers' comments.

The reviewers collectively indicate a need for clarification of some of the technical details of the model and better justification of/additional context for the simplifying assumptions. Further, Reviewer 1 in particular suggests contextualizing "diversity" as used in this model against diversity as measured in real data, which will facilitate interpretation of these results in a real-world context and should improve the reach of the paper.

We cannot make any decision about publication until we have seen the revised manuscript and your response to the reviewers' comments. Your revised manuscript is also likely to be sent to reviewers for further evaluation.

Sincerely,

Nic Vega, Ph.D.

Academic Editor

PLOS Computational Biology

James O'Dwyer

Section Editor

PLOS Computational Biology

The reviewers collectively indicate a need for clarification of some of the technical details of the model and better justification of/additional context for the simplifying assumptions. Further, Reviewer 1 in particular suggests contextualizing "diversity" as used in this model against diversity as measured in real data, which will facilitate interpretation of these results in a real-world context and should improve the reach of the paper.

Reviewer's Responses to Questions

**Comments to the Authors:**

Reviewer #1: Please see attached pdf.

Reviewer #2: This paper provides one scenario in which biological populations maintain their diversity through a metacommunity composed of patches against the mechanism of genetic diversity maintenance by neutral mutation and the effect of diversity reduction caused by gene-specific sweeps associated with beneficial horizontal gene transfer (HGT.)

Using numerical simulations and mathematical models, the authors showed that the neutral transfer of individuals or genes among patches and the fixation of beneficial genes could restore diversity, provided that the transfer rate is on a reasonable timescale.　The authors argue that this allows diversity to be maintained without assuming high rates of HGT, as claimed in some papers.

The conclusion is not far-fetched and seems reasonable. However, I think it should be added, "Under some assumptions in this study...".

I have some questions, which I will list below.

1. the periodicity of the fixation time of the beneficial gene is questionable. In nature, it is acceptable even if the mean of τ_fix is given by a statistic such as Eq. 13, but it seems to occur randomly. So, shouldn't we insist that diversity is not zero in the long-time average?

2. Is the total number of individuals in this model conserved?　The maximum value of B(t) is M, but since D(t) decreases independently, the number of individuals fluctuates, affecting the diversity.

3. 3. is the limit in Eq. 25 consistent with Eq. 2?

Minor remarks

P. 12 tau_fix -> τ_fix

P. 20 Delete (23).

Reviewer #3: For your convenience, I have uploaded my review as a PDF (to access URLs linking additional references) and in Word doc form (to make it easier to write responses to my comments).

I sincerely thank both the editor and the authors for the opportunity to review such an interesting and relevant manuscript.

Reviewer #4: I have reviewed the manuscript "A Metacommunity Scenario Preserves the Diversity in Presence of Gene-specific Selective Sweeps". It explores the context of horizontal evolution in biological communities, indicating how metapopulation dynamics (explicit space) contribute to the maintenance of general biodiversity in communities like bacteria, which generates low rates of genetic variability. The authors employ stochastic dynamics to represent the problem and assume that the chances of migration, innovation, and fixation are low compared to the number of individuals and the system as a whole. In comparison, the authors demonstrate the behavior of neutral dynamics that gradually gains new elements, such as innovation and horizontal gene transfer.

Accordingly, the emergence of a beneficial gene generates strong positive selection pressure, reducing biodiversity by positive selection, while horizontal gene transfer (HGT) of benefic genes to other species/strains contribute to the maintenance of species richness. The authors explore how the rate of new gene generation relative to transfer affects the diversity of the metacommunity context. The simulations demonstrate that the rates of new benefic genes reduce the total biodiversity, while the horizontal transfer of benefic genes contributes to the maintenance of species/strains diversity. Further, it is demonstrated to exist a cyclical behavior of the total number of species when both new strains and HGT interact. By tuning the parameters, it is demonstrated that the total biodiversity is modulated by relative innovation rates.

The article is well-written. The ideas are organized and conveyed clearly and cohesively throughout the text. The authors focused mainly on how low HGT rates are capable of contributing to the maintenance of biodiversity due to interaction with spatial dynamics (migration events). Indeed, the results demonstrate that even low HGT rates produce higher rates of biodiversity than those explained by neutral dynamics or selection, an effect resulting from the metapopulation context. Therefore, the paper presents an interesting proof of concept and plausibility regarding the role of HGT and in the maintenance of species diversity, even under low rates of horizontal transfer, at least from a qualitative point of view.

Despite the relevance of the presented topic and the paper has achieved the main objective, there are a few minor weaknesses. The article's results have the premise that the rates of emergence of new beneficial strains and their fixation scale with the migration rate. Although the parameters presented by the article connote different phenomena from other models found in the literature, the underlying process has already been explored by multispecies models in related areas. In all cases, conclusions suggest that the effect of spatial dynamics on the maintenance of multispecies populations mediates the antagonistic forces, in terms of rates. Therefore, the model, by itself, represents a new application of stochastic dynamics more than an innovation by itself.

In addition, the boundary conditions of the problem arise doubts concerning the role of HCT to maintain species diversity in more intricate ecological-evolutionary systems. This topic is indeed recognized by authors, and it relativizes whether results achieved by simulations are really applicable to real-world systems.

Finally, terms like population and species are sometimes employed in sense of statistical modeling, while sometimes they are interchangeably used in the sense of biology. Because the article deals with both aspects, some passages may lead the most inattentive reader to misinterpret the intended idea. Therefore, I suggest that the authors review the use of these terms throughout the text as a way to provide greater clarity.

**Have the authors made all data and (if applicable) computational code underlying the findings in their manuscript fully available?**

Reviewer #1: None

Reviewer #2: Yes

Reviewer #3: Yes

Reviewer #4: Yes

PLOS authors have the option to publish the peer review history of their article (what does this mean?). If published, this will include your full peer review and any attached files.

Reviewer #1: **Yes: **Zachary R Miller

Reviewer #2: No

Reviewer #3: No

Reviewer #4: No
---

## [Decision Letter · Decision Letter 1]

13 Aug 2023

Dear Dr. Pompei,

Thank you very much for submitting your manuscript "Metacommunity Structure Preserves Genome Diversity in the Presence of Gene-specific Selective Sweeps Under Moderate Rates of Horizontal Gene Transfer." for consideration at PLOS Computational Biology. As with all papers reviewed by the journal, your manuscript was reviewed by members of the editorial board and by several independent reviewers. The reviewers appreciated the attention to an important topic. Based on the reviews, we are likely to accept this manuscript for publication, providing that you modify the manuscript according to the review recommendations.

Sincerely,

James O'Dwyer

Section Editor

PLOS Computational Biology

James O'Dwyer

Section Editor

PLOS Computational Biology

Reviewer's Responses to Questions

**Comments to the Authors:**

Reviewer #1: See attached comments.

Reviewer #2: The authors responded appropriately to my questions and suggestions.I have nothing new to point out.

Reviewer #3: Please see the attached word document and PDF.

**Have the authors made all data and (if applicable) computational code underlying the findings in their manuscript fully available?**

Reviewer #1: Yes

Reviewer #2: Yes

Reviewer #3: Yes

PLOS authors have the option to publish the peer review history of their article (what does this mean?). If published, this will include your full peer review and any attached files.

Reviewer #1: **Yes: **Zachary R Miller

Reviewer #2: No

Reviewer #3: No

Figure Files:

Data Requirements:

Reproducibility:

References:

---

## [Editor Report · Decision Letter 2]

19 Sep 2023

Dear Dr. Pompei,

We are pleased to inform you that your manuscript 'Metacommunity Structure Preserves Genome Diversity in the Presence of Gene-specific Selective Sweeps Under Moderate Rates of Horizontal Gene Transfer.' has been provisionally accepted for publication in PLOS Computational Biology.

Best regards,

James O'Dwyer

Section Editor

PLOS Computational Biology

James O'Dwyer

Section Editor

PLOS Computational Biology

---

## [Editor Report · Acceptance letter]

27 Sep 2023

PCOMPBIOL-D-23-00075R2 

Metacommunity Structure Preserves Genome Diversity in the Presence of Gene-specific Selective Sweeps Under Moderate Rates of Horizontal Gene Transfer.

Dear Dr Pompei,

I am pleased to inform you that your manuscript has been formally accepted for publication in PLOS Computational Biology. Your manuscript is now with our production department and you will be notified of the publication date in due course.

With kind regards,

Anita Estes
